# Development and Characterization of Pickering Emulsion Stabilized by Walnut Protein Isolate Nanoparticles

**DOI:** 10.3390/molecules28145434

**Published:** 2023-07-15

**Authors:** Jiongna Liu, Hengxuan Zhang, Xue Sun, Fangyu Fan

**Affiliations:** 1College of Life Sciences, Southwest Forestry University, Kunming 650224, China; 2Key Laboratory for Forest Resources Conservation and Utilization in the Southwest Mountains of China, Ministry of Education, Southwest Forestry University, Kunming 650224, China; 3Key Laboratory of Forest Disaster Warning and Control of Yunnan Province, Kunming 650224, China; 4Key Laboratory of National Forestry and Grassland Administration on Biodiversity Conservation in Southwest China, Southwest Forestry University, Kunming 650224, China

**Keywords:** walnut protein isolates, nanoparticle, Pickering emulsion, stability

## Abstract

This study was conducted to prepare walnut protein isolate nanoparticles (nano-WalPI) by pH-cycling, combined with the ultrasound method, to investigate the impact of various nano-WalPI concentrations (0.5~2.5%) and oil volume fractions (20~70%) on the stability of Pickering emulsion, and to improve the comprehensive utilization of walnut residue. The nano-WalPI was uniform in size (average size of 108 nm) with good emulsification properties (emulsifying activity index and stability index of 32.79 m^2^/g and 1423.94 min, respectively), and it could form a stable O/W-type Pickering emulsion. When the nano-WalPI concentration was 2.0% and the oil volume fraction was 60%, the best stability of Pickering emulsions was achieved with an average size of 3.33 μm, and an elastic weak gel network structure with good thermal stability and storage stability was formed. In addition, the emulsion creaming index value of the Pickering emulsion was 4.67% after 15 days of storage. This study provides unique ideas and a practical framework for the development and application of stabilizers for food-grade Pickering emulsions.

## 1. Introduction

Emulsions stabilized by solid particles are referred to as Pickering emulsions (PEs) [1]. Compared with traditional emulsions of surfactants as an emulsifier, PEs have many advantages, including excellent stability, a low toxic nature, cheap production cost, and an environmentally friendly nature [2,3]. Solid particles can be inorganic or organic particles. However, the utilization of most inorganic particles (graphene [4], Fe_3_O_4_ [5], and SiO_2_ [6]) was limited in the food industry or pharmacy because of their toxicity. Polysaccharides, proteins, and fats of food-grade organic particles can replace inorganic particles as emulsifiers to prepare PEs, which have several potential utilities in the field of medicine, cosmetics, and food [7]. Compared with other organic particles, proteins have a good hydrophilic/oleophilic balance, and they are biocompatible and degradable; proteins can also form oil-in-water (O/W), water-in-oil (W/O), and multiple emulsions, and O/W emulsion are predominant [8]. Moreover, proteins can be used as a nutritional supplement to enhance the food value of emulsions. Plant proteins can be employed to stabilize PEs, and they have a wide source, low price, and fewer allergens when compared with animal proteins. The high gluten content of most plant proteins leads to poor water solubility and emulsification, which limits their application in the food field [9]. Previous studies have shown that plant proteins can be prepared as nanoparticles wit a regular shape and uniform size by using the pH-cycling method, and nano-proteins can improve the solubility and emulsification of plant proteins and prepare stable PEs as an emulsifier [10,11,12]. Lee et al. [11] found that ultrasonication combined with the pH-cycling method (US–pH-cycling method) can be used to achieve a smaller particle size and turbidity and higher solubility and surface hydrophobicity (H_0_) of soy protein isolate compared to a pH-cycling treatment. The protein structure unfolded at a pH of 12.0, and ultrasound cavitation caused fragmentation of the protein and a decrease in particle sizes. Next, the pH was adjusted to neutral, and the protein refolded to form nanoparticles [11].

Walnut (*Juglans regia* L.) is a great nutritional nut because of its high levels of unsaturated oils, proteins, polyphenols, vitamins, and other nutrients, and it has several potential health benefits, including antioxidant properties, immune regulation, and cholesterol reduction [13,14]. The walnut residue is the byproduct of walnut processing, with a protein content of more than 50% [15]. However, given the high level of insoluble protein, the comprehensive utilization rate of the walnut meal is low, resulting in a serious loss of protein resources. Walnut protein isolate (WalPI) has amphiphilic and adjustable structures, and it is an ideal substitute for biopolymers [16]. Therefore, the walnut protein isolate nanoparticles (nano-WalPI) were prepared by the US–pH-cycling method to improve the comprehensive utilization rate of the walnut meal, and PEs were prepared using nano-WalPI as an emulsifier.

As a thermodynamically unstable system, the traditional emulsion is susceptible to some adverse phenomena, including aggregation, flocculation, sedimentation, and coalescence [17]. The stability of the PE system depends significantly on the property of the emulsifier (nanoparticle). The stability of the PEs by nano-protein is impacted by internal and external parameters, including the properties and concentration of nano-protein, the oil volume fraction, and the pH [18]. Sun et al. [10] discovered that PEs using a ternary nano complex (zein, sodium caseinate, and propylene glycol alginate) demonstrated “solid-like” properties as the oil volume fraction increased (50–80%), but stability deteriorated when the oil volume fraction reached 81%. Zhu et al. [7] investigated the influence of oil volume fractions on the stability of PEs via complex particles (zein and corn fiber gum) and found that the particle size, viscosity, and stability of the emulsion increased as oil volume fractions increased. Li et al. [19] discovered that the droplet size and emulsion creaming index (CI) of PEs were impacted by the concentration of zein/gum Arabic particles (ZGPs) and oil volume fractions, and the stability of PEs was enhanced as the concentration of ZGPs or oil volume fraction increased. However, as the oil volume fraction increases, the droplet size of the PEs increases, but their CI value decreases. This is due to the increase of oil volume fractions, which enhances the packing density and extent of interaction among PEs droplets and improves the gel strength and stability of the PEs [7].

In the current study, nano-WalPI, a new PE emulsifier, was prepared by using the US–pH-cycling method, which was then employed to prepare PEs. The particle size, microstructure, polydispersity index (PDI), H_0_, secondary structure, and emulsion properties of nano-WalPI were studied to comprehensively understand particle properties. In addition, the effect of nano-WalPI concentrations and oil volume fraction on the droplet size, rheological properties, thermal stability, and storage stability of PEs was studied. This research may expand the application of walnut protein in the field of food and provide an alternative stabilizer for PEs.

## 2. Results and Discussion

### 2.1. Nano-WalPI Characterization

#### 2.1.1. Analysis of the Physicochemical Properties of WalPI and Nano-WalPI

The stability of PEs may be directly affected by nanoparticle size, and decreasing particle size may improve the capacity of solid particles to adsorb at the interface [20]. As shown in Figure 1A,B, the size distribution of WalPI and nano-WalPI showed a single peak that indicated the existence of a monodisperse system, which contributes to the formation of a stable emulsion system. The size distribution of WalPI and nano-WalPI ranges from 1990 to 4200 nm, as well as from 24 to 220 nm, with average sizes of 3368 and 108 nm, respectively (Table 1). The difference in particle size led to the poor solubility of WalPI, the precipitation of the solution, the good solubility of nano-WalPI, a uniform solution, and no precipitation (Figure 1A). In addition, the nano-WalPI was small, and it had a better PDI (0.40) compared to WalPI (0.56). The results indicated that the US-pH-cycling method could reduce the size and PDI value of WalPI. Under extremely alkaline conditions, strong electrostatic repulsion among WalPI molecules led to a partial unfolding of the WalPI structure. At pH 7.0, the intermolecular electrostatic repulsion of WalPI decreased; the quaternary structure of WalPI was disrupted, and WalPI molecules refolded to a certain extent [9]. Meanwhile, the ultrasonic action promotes the unfolding of the WalPI structure, weakens non-covalent interactions of WalPI intermolecularly, and blocks WalPI aggregation because of the cavitation effect and mechanical shear. The results changed the average size of WalPI and strengthened the interaction of the protein with water, which enhanced its solubility and emulsifying ability [21,22].

The H_0_ reflects the distribution of hydrophobic groups on the protein surface, and its size is associated with the emulsifying properties [21]. As shown in Table 1, compared with WalPI, the H_0_ of nano-WalPI increased significantly from 875.21 to 1027.43 (*p* < 0.05). The reason was that the unfolding and re-folding of the protein during pH treatment led to the relocation of protein hydrophobic groups [11]. Meanwhile, the ultrasonic action causes the protein to unfold and break off, exposing the hydrophobic groups of the protein, thus increasing the H_0_ of the protein [22].

#### 2.1.2. Scanning Electron Microscopy

Figure 1C displays the SEM images of nano-WalPI. WalPI and nano-WalPI were spherical with a smooth surface. The sizes of WalPI and nano-WalPI were approximately 3000 and 150 nm, respectively. The results indicated that the US–pH-cycling method could reduce the size of WalPI to prepare spherical nanoparticles with uniform sizes, which was similar to the findings shown in Section 2.1.1. This result was primarily due to the uniform distribution of hydroniums by D-glucono-delta-lactone (GDL) hydrolysis during the preparation of nano-WalPI, which induces proteins to form abundant and homogeneous nanoparticles [23]. Moreover, ultrasound treatment can generate cavitation activities, which can break the protein aggregates and reduce the size of proteins [24].

#### 2.1.3. FTIR Analysis

The functional group of WalPI and nano-WalPI was analyzed by Fourier transform infrared spectroscopy (FTIR). Moreover, we calculated protein secondary structure contents using the amide I band (1600–1700 cm^−1^) of the protein [25]. As shown in Figure 1D, the characteristic peak of WalPI and nano-WalPI at 3313.3 and 3404.5 cm^−1^ is attributed to stretching vibrations of OH, and that at 2965.4 and 2931.4 cm^−1^ to stretching vibrations of CH [23]. The absorption peaks of amide I (1600~1700 cm^−1^) and the amide II bands (1500~1600 cm^−1^) are attributed to the stretching vibrations of C=O and bending vibrations of N-H, respectively [13]. Additionally, the main absorption peaks for WalPI (nano-WalPI) were from OH bending at 1401 (1389.6) cm^−1^, C-N stretching, and N-H bending at 1241 (1236.2) cm^−1^. The peak position of nano-WalPI was blue-shifted in different degrees in comparison to the FTIR spectra of WalPI, and a new peak emerged at 1066.2 cm^−1^ that was attributed to the stretching of C-O. This finding demonstrated that the US–pH-cycling method might encourage the unfolding or dissociation of the protein structure and then change protein conformation [11]. As can be seen from Table 2, the β-sheet of nano-WalPI decreased from 53.67% to 39.67% compared with the WalPI, whereas the β-turn, α-helix, and random coil increased. -OH and -COOH may have caused a decrease in the β-sheet and transformed its structure into α-helix and β-turn [25,26]. In addition, the protein structure unfolds under alkaline conditions, and the ultrasound treatment disrupted the hydrogen bonds, which resulted in the change from an ordered structure to a disordered structure [26]. A similar pattern of results was discovered by Zhu et al. [27], who found that the ultrasound treatment (ranging from 200 to 600 W) reduced the particle size (2760–210 nm) of WalPI and changed its solubility and secondary structure.

#### 2.1.4. Emulsion Type Analysis

As shown in Figure 1E, the PEs were O/W types because WalPI- and nano-WalPI-stabilized emulsion drops were agglomerated into droplets in the oil phase and uniformly dispersed in the water phase. However, some oil was observed in the upper layer of WalPI-stabilized PEs, indicating that the WalPI cannot form stable PEs because of the poor emulsion stability caused by the large WalPI particle size.

#### 2.1.5. Emulsifying Properties

The emulsifying activity index (EAI) and stability index (ESI) of a protein refer to its ability to rapidly adsorb to the oil–water interface to prevent flocculation and precipitation during emulsion formation and its adsorption capacity on the oil–water interface [28], respectively. Figure 1F displays the EAI and ESI of WalPI and nano-WalPI, respectively. Compared with WalPI, the EAI and ESI of nano-WalPI were 32.79 m^2^/g and 1423.94 min, respectively, increasing by 2.54 and 4.72 times, indicating that the US–pH-cycling method may enhance the interaction of proteins and oil. The unfolding of the protein structure improved the flexibility and surface hydrophobicity of the nano-WalPI [29] and enhanced the adsorption rate of the nano-WalPI to the oil–water and gas–liquid interfaces, increasing EAI and ESI of the nano-WalPI. Moreover, the small diameter of the protein increased its solubility and adsorption on the interface and increased the stability of emulsions [24,30], which was consistent with the abovementioned results. Therefore, the nano-WalPI was utilized as an emulsifier to produce PEs, and the stability of PEs was estimated by changing the nano-WalPI concentration and oil volume fraction.

### 2.2. Effects of Nano-WalPI Concentrations on PEs

#### 2.2.1. PE Droplets

As shown in Figure 2A,B, the average size of the PEs steadily reduced as the concentration of nano-WalPI was enhanced, and droplet-size-distribution profiles of all PEs displayed a single, narrowly shaped peak. At a 0.5% concentration of nano-WalPI, the PEs had the largest average size, measuring 23.79 μm, but their droplet size distribution was wide, indicating that the uniformity of the emulsion was poor. By contrast, the average size of the PEs significantly decreased by 89.00% to 3.33 μm (*p* < 0.05), and their droplet size distribution narrowed when the nano-WalPI concentration increased to 2.0%. The oil–water interface is sufficiently filled with solid particles as the concentration of nano-WalPI increases, which results in smaller droplets [31]. When the nano-WalPI concentration exceeded 2.0%, the droplet size of the PEs remained unchanged, indicating that the nano-WalPI concentration did not influence the droplet size of the PES. The adsorption of nano-WalPI at a certain concentration was limited in the oil–water interface by steric hindrance and electrostatic repulsion, resulting in a constant droplet size [32].

#### 2.2.2. Rheological Properties

Figure 2C,D show the influence of nano-WalPI concentrations on the rheological characteristics of PEs. The apparent viscosity of the PEs reduced from 15.20 Pa·s to 0.19 Pa·s at a shear rate of 0.1 s^−1^ as nano-WalPI concentrations were increased (0.5–2.0%). The nano-WalPI adsorption of the oil–water interface reached saturation; the stability of PEs increased, and apparent viscosity decreased as the nano-WalPI concentration increased to a certain level. At shear rates of 0.1 to 60 s^−1^, PEs with nano-WalPI concentrations of 0.5% to 1.5% exhibited shear-thinning behavior, as rising shear rates resulted in a reduction in the apparent viscosity of PEs. The network structure of the PEs was broken or rearranged under shear behavior [31], where the emulsion destruction rate exceeded the reorganization rate. In addition, at a shear rate of 60–100 s^−1^, the apparent viscosity of PEs remained unchanged, and they exhibited Newtonian fluid behavior [33]. This phenomenon may be attributed to the result that the disruption rate of the emulsion is consistent with the recombination rate [34]. As the nano-WalPI concentration increased (2.0–2.5%), the apparent viscosity of PEs increased (0.19–0.21 Pa·s) at a shear rate of 0.1 s^−1^. While the thickness of the interfacial layer exceeded the critical value, excess nano-WalPI was repelled by the emulsified droplets, causing the PEs to flocculate [35,36]. Furthermore, because the excess nano-WalPI wasn’t uniformly dispersed [13], the nano-WalPI combined to form insoluble complexes, resulting in the flocculation of the droplet and an increase in the apparent viscosity of the PE. The PEs with nano-WalPI concentrations of 2.0% to 2.5% exhibited shear-thinning behavior at a shear rate of 0.1–10 s^−1^ and Newtonian fluid behavior at a shear rate of 10–100 s^−1^. The reason is that the solid particles adsorbed on the droplet surface may form structured interfacial layers, which may affect the overall rheology of the PEs [36]. The shear stress of PEs increased with the shear rate. The shear stress of the PEs decreased and then increased with increasing nano-WalPI concentrations (0.5–2.5%) at a shear rate of 0.1 s^−1^, which was similar to the ordering of its apparent viscosity.

Figure 2E illustrates that the storage modulus (G′) and loss modulus (G″) were used to express the dynamic modulus of PEs. At a frequency of 0.1 Hz, the G′ and G″ values of PEs decreased and then increased with increasing nano-WalPI concentration (0.5–2.5%), reaching their maximum at 0.5% of the nano-WalPI concentration, which was primarily attributed to the lack of nano-WalPI to fully cover an interface, resulting in irreversible bridging flocculation [37]. At a frequency of 0.1–10 Hz, all PEs exhibited “solid-like” properties of elastic deformation. At a frequency of 10–100 Hz, the properties of PEs were predominantly viscous (G′ < G″), with a nano-WalPI concentration of 1.0%, 1.5%, and 2.5%, showing that the gel structure of the PEs was disrupted, whereas the PEs with nano-WalPI concentrations of 2.0% showed an elastic weak gel network structure (G′ > G″), indicating the existence of an elastic gel structure [38]. The gel structure of the PEs can prevent droplet agglomeration and improve stability. Thus, the PEs are more stable at 2.0% of nano-WalPI concentration.

#### 2.2.3. Thermal Stability

Heat treatment was an essential method for food processing. Protein-stabilized O/W emulsions were sensitive to high temperatures, and the emulsion system was subject to flocculation, aggregation, and phase separation. Figure 2F shows that the PEs for a nano-WalPI concentration of 0.5% was severely stratified after heating in a boiling water bath, whereas the PEs for a nano-WalPI concentration of more than 0.5% was not stratified before or after heat treatment, which indicated that the thermal stability of the PEs improves as nano-WalPI concentration increases. Considering that the low concentrations were loosely arranged on the oil–water interface, oil droplets gathered together, resulting in the poor thermal stability of PEs [19]. At a nano-WalPI concentration higher than 1.5%, a decrease in the flowability of PEs was observed by heat treatment, indicating that heat treatment improved the stability of PEs. Studies have shown that heat treatment can expose protein hydrophobic regions [39], and protein structures are unfolded, which facilitates the ordered rearrangement of protein molecules and leads to enhanced protein amphiphilicity and the capacity of adsorption to interfaces. In addition, heat treatment leads to the denaturation of proteins and forms more network structures, thereby improving emulsion stability [21].

#### 2.2.4. Storage Stability

The morphology and CI value of PEs at different nano-WalPI concentrations during storage are shown in Figure 2G and Table 3. The emulsified layer of PEs dropped with longer storage time and increased with the increase of nano-WalPI concentration. PEs with a nano-WalPI concentration of 0.5% had the highest CI value, reaching 18.07% after 15 days of storage, because the interfacial electrostatic interaction weakened, and the nano-WalPI could not stabilize the oil–water interface with the increase of storage time. Meanwhile, gravity and Brownian motion caused the oil droplets to aggregate and PEs to delaminate [39]. The CI value of PEs gradually reduced with the increase of nano-WalPI concentration; the smallest CI value was observed at the nano-WalPI concentration of 2.0% after 15 days of storage time, and a significant decrease (74.16%, *p* < 0.05) was observed when nano-WalPI concentrations were enhanced from 0.5% to 2.0%. The amount of nano-WalPI adsorbed at the oil–water interface increased, and the interface film thickened, which prevented oil droplet aggregation. In addition, sufficient nano-WalPI can slow down the movement of emulsions, form a stable network structure, and promote the stability of PEs [40].

### 2.3. Effects of Oil Volume Fraction on PEs

#### 2.3.1. PE Droplets

According to the results in Section 2.2, PEs were the most stable at a nano-WalPI concentration of 2.0%. Therefore, the stability of PEs with different oil volume fractions was investigated at a nano-WalPI concentration of 2.0%. Figure 3A,B display the average size and size distribution of PEs. The distribution of PE droplet size showed a single peak system at all oil volume fractions samples, which was the monodisperse system. The average size of PEs increased (3.07–8.95 μm) significantly (*p* < 0.05) as the oil volume fraction increased (20–70%). The area of the oil–water interface expands as the oil volume fraction increases, and nano-WalPI is not enough to cover the area, which causes droplets to collide, aggregate, and tightly stack, thereby increasing the average size of PE droplets [7].

#### 2.3.2. Rheological Properties

Figure 3C,D illustrate the impact of oil volume fractions on the apparent viscosity and shear stress of PEs. At a shear rate of 0.1 s^−1^, the apparent viscosity of PEs increased (0.03–2.72 Pa·s) with the increase of oil volume fraction (20–70%), whereas the apparent viscosity of PEs at 20% oil volume fraction was 30 times that of water (0.001 Pa·s) [41]. More oil droplets were absorbed by nano-WalPI, creating a stable three-dimensional network structure [19]. The PEs with oil volume fractions of 20–50% exhibited shear-thinning behavior at a shear range of 0.1–5 s^−1^, probably because the randomly distributed droplets were arranged orderly along the streamlined direction in the flow velocity field with increased shear rate, leading to the reduction of friction among droplets [42]. PEs showed Newtonian fluid properties at a shear range of 5–100 s^−1^. The PEs with 60% and 70% oil volume fractions displayed shear-thinning behavior at shear ranges of 0.1–10 s^−1^ and 0.1–50 s^−1^, respectively, whereas they showed Newtonian fluid behavior at a shear range of 10–100 s^−1^ and 40–100 s^−1^, respectively. At high oil volume fractions, the PE system had a relatively low nano-WalPI content and tightly connected droplets, which reduced its fluidity [7]. At a shear range of 0.1–100 s^−1^, the shear stress of PEs increased with the increase of oil volume fraction (20–70%) and shear rate. Excessive oil droplets might enhance the collision among droplets. Given the interaction between oil and proteins, the close connection among proteins strengthened the density of the PE gel network as the oil volume fraction increased [43], leading to a rise in the apparent viscosity and shear stress of PEs.

As shown in Figure 3E, the G′ and G″ values of PEs increase with the increase of frequency, indicating that they displayed frequency dependence. At a frequency of 0.1 Hz, the G′ and G″ values of PEs gradually increased with the increase of the oil volume fraction, indicating that oil droplets play a role as an “active filler” in PEs [44], and the stability of the oil–water interfacial adsorption layer of PEs is improved [19]. At high oil volume fractions, the protein content is relatively low in the continuous phase, and a large number of oil droplets promotes the rearrangement of the gel network and participates in the formation of the network structure due to the electrostatic repulsion and merging of the oil droplets. In the initial frequency range (0.1–10 Hz), the PEs for an oil volume fraction of 20–60% displayed “solid-like” properties of elastic deformation (G′ > G″), and, as the frequency increased (10–100 Hz), the G′ value remained greater than G″, demonstrating that PEs formed an elastic gel network structure. The PEs with a 70% oil volume fraction showed a change in the rheological behavior from predominant viscosity (0.1–5 Hz, G′ < G″) to predominant elasticity (5–100 Hz, G′ > G″), showing that it contained a weak gel structure. When the oil volume fraction is low, the aggregation and cross-linking effect of nano-WalPI is enhanced. This finding hindered the accumulating and coalescing of oil droplets, and the dispersed oil droplets could not provide integration power for the gel system [45]. Therefore, rising oil volume fractions could improve the gel network structure and stability of PEs.

#### 2.3.3. Thermal Stability

The morphology of PEs at various oil volume fractions is displayed in Figure 3F before and after heat treatment. PEs with varying oil volume fractions have no stratification phenomenon before or after heat treatment. We also found that the viscosity of PEs increased after heating as the oil volume fraction increased, and their stability improved probably because the oil droplet can be regarded as the “active filler” of the PE gel network; increased oil volume fractions also reinforced the density of the gel network structure and stability of PEs [45]. In addition, heating caused the protein disulfide bonds to break, exposing the active site of the protein and enhancing protein–protein interactions, which improved the viscosity and storage stability of PEs [46,47]. The findings demonstrated that the PEs exhibited excellent thermal stability at various oil volume fractions and that stability increased with oil volume fractions.

#### 2.3.4. Storage Stability

The morphology and CI value of PEs at different oil volume fractions during storage are displayed in Figure 3G and Table 4. The emulsified layer of PEs reduced with a rise in storage time and increased with a rise in oil volume fraction. During storage, the PEs with a 20% oil volume fraction had the highest CI value, reaching 91.05% after 15 days. The CI value of PEs significantly reduced from 91.05% to 7.17% after 15 days (*p* < 0.05), as oil volume fractions were enhanced (20–70%). The storage stability of PEs was improved by the high oil volume fraction because of a high filler density and viscosity of the emulsion as well as the slow rate of droplet migration [48]. The PEs had the smallest CI value when the oil volume fraction was 60%, and the CI value was 4.67% after 15 days. PEs form a gel network structure that traps the emulsified oil droplets, limiting droplet coalescence [49]. The high volume fraction might increase the collision among droplets when the oil volume fraction exceeds 60% [43]. Li et al. [19] showed that raising the oil volume fraction (10–60%) could increase the apparent viscosity and stability of PEs, which was compatible with our findings.

## 3. Materials and Methods

### 3.1. Materials

The walnut residue (cold-pressed) was purchased from Qiaojia Tuwei Food Co., Ltd. (Zhaotong, China). *Camellia oleifera* seed oil (COSO) was acquired from Yihai Kerry Investment Co., Ltd. (Shanghai, China). D-glucono-delta-lactone (GDL) was acquired from Mclean Biochemical Technology Co., Ltd. (Shanghai, China). 8-Anilino-1-naphthalenesulfonic acid (ANS) was purchased from Shanghai Source Leaf Biological Technology Co., Ltd. (Shanghai, China). The chemical reagents used in the test were of analytical grade.

### 3.2. Walnut Protein Isolate Preparation

WalPI was prepared in accordance with the methodology of Zhang et al. [14]. The composition of WalPI is as follows: 80.17% protein, 2.22% ash, 7.83% water, and 0.87% fat.

### 3.3. Nano-WalPI Preparation

Nano-WalPI was prepared by using the US–pH-cycling method; the operation process was in accordance with the method of Lee et al. [11] with some minor adjustments. WalPI 2.0% (*w*/*v*) was added to distilled water, and a 1.0 mol/L NaOH solution was utilized to adjust the pH of the protein solution to 11.5. After stirring at 200 r/min for 4 h at room temperature, the protein solution was sonicated by using an ultrasonic apparatus (Xinyi-IID, Ningbo, China) for 5 min in an ice bath (570 w, 25 kHz). After standing for 1 h, the protein solution was stirred at 200 r/min while adding 10.0% (*w*/*v*) GDL solution to adjust the pH to 7.0. Next, the protein solution was centrifuged for 15 min at 4 °C and 12,000 r/min (5804R Centrifuge, Eppendorf, Darmstadt, Germany). The supernatant was freeze–dried after being dialyzed (molecular weight cutoff, 3500) for 48 h against distilled water. The recovery of protein was 80.45%.

### 3.4. Nano-WalPI Characterization

#### 3.4.1. Particle Size and PDI

Using a Zetasizer Nano ZS (Malvern, UK), a 2.0% (*w*/*v*) nano-WalPI solution was measured at 25 °C to obtain the particle size distribution, average size, and PDI.

#### 3.4.2. Surface Hydrophobicity

The H_0_ of the WalPI and nano-WalPI was determined in accordance with the method of Zhang et al. [14]. The sample was diluted to 0.005, 0.01, 0.02, 0.05, 0.1, 0.2, and 0.5 mg/mL with 0.01 mol/L phosphate buffer (pH = 7.0). The sample solution of 2 mL was mixed with 40 μL of ANS solution and left for 15 min at 25 °C. The fluorescence intensity was measured at a 390 nm excitation wavelength and 470 nm emission wavelength with a slit of 5 nm. The protein concentration and fluorescence intensity were plotted as the abscissa and vertical coordinates; the slope of the curve is H_0_.

#### 3.4.3. SEM Analysis

The microstructure of WalPI and nano-WalPI was observed by using a scanning electron microscope (Sigma 300, ZEISS, Oberkochen, Germany) at an accelerating voltage of 3 kV, and the magnification was 10,000× *g*.

#### 3.4.4. FTIR Analysis

The samples were prepared using the potassium bromide procedure, and infrared spectra were collected using FTIR (IRPrestige-21, Shimadzu, Kyoto, Japan). The scanning range was 4000–400 cm^−1^, with 32 scans and a resolution of 4 cm^−1^. The relative content of the secondary structure was calculated by using Peak Fit 4.12 and OMNIC 8.2 [14].

#### 3.4.5. Analysis of Emulsion Types

COSO (8 mL) and distilled water (8 mL) each received 1 mL of emulsion. Whenever the emulsion drop disperses in water, it is identified as O/W; otherwise, it is W/O [19].

#### 3.4.6. Emulsifying Properties

Based on the approach of Li et al. [20], the *EAI* and *ESI* were used to estimate the emulsifying capabilities of nano-WalPI. Eight milliliters of a nano-WalPI solution (*C*, 2.0%, *w*/*v*, pH 8.0) was dispersed using a high-speed disperser (FJ200-SH, Wuxi, China) at 12,000 r/min for 2 min. Then, 12 mL of COSO (*φ*, 60%) was added slowly to the mixed solution and dispersed at 12,000 r/min for 3 min. Next, the emulsion was prepared by ultrasonically dispersing the mixed solution for 2 min in an ice bath (475 W, 25 kHz).

Emulsions (100 μL) were obtained immediately after preparation, after being diluted 300 times with 0.1% (*w*/*v*) SDS (*N*, 300). A UV–vis spectrophotometer (UV-2600, Shimadzu, Japan) was used to measure the absorbance of diluted emulsions at 500 nm at 0 (*A*_0_) and 30 (*A*_30_) min with 0.1% SDS as the blank [20]. Equations (1) and (2) were utilized to calculate the EAI and ESI.
(1)$$EAI(m^{2}/g)=2\times 2.303\times \frac{A_{0}\times N}{C\times \varphi \times 10000}$$
(2)$$ESI(\min)=\frac{A_{0}}{A_{0}-A_{30}}\times 30$$
where *A*_0_ and *A*_30_ are the absorbance values of the dilution emulsion at 0 and 30 min; *N* is the dilution ratio; *C* is the protein concentration; and *φ* is the oil volume fraction.

### 3.5. PE Preparation with Different Nano-WalPI Concentrations and Oil Volume Fractions

The PEs with different nano-WalPI concentrations (0.5%, 1.0%, 1.5%, 2.0%, and 2.5%, *w*/*v*) were prepared at the same oil volume fraction (60%). Meanwhile, the PEs were prepared at a 2.0% nano-WalPI concentration using different oil volume fractions (20%, 30%, 40%, 50%, 60%, and 70%). During the preparation of the PEs, the mixed solution was homogenized for 3 min at 12,000 r/min to create the PEs after the oil was completely mixed.

### 3.6. Characterization of PEs

#### 3.6.1. Droplet Size

The droplet and average size of the PEs were determined by using a laser scattering particle size distribution analyzer (LA-960V2, Horiba, Kyoto, Japan) at 25 °C [14].

#### 3.6.2. Rheological Properties

At 25 °C, a rheometer was applied to assess the rheological properties of PEs. The apparent viscosity and shear stress of PEs were recorded when the shear rate was increased by 0.1–100 s^−1^. The G′ and G″ values were determined at a constant strain of 0.5% in a frequency range of 0.1–100 Hz, and the dynamic viscoelasticity of PEs was characterized.

#### 3.6.3. Thermal Stability

The thermal stability of PEs was estimated in accordance with the method of Li et al. [21]. The PEs in a volume of 5 mL were heated for 0.5 h in boiling water before being cooled to 25 °C. The macroscopic morphology of PEs was investigated before and after heating by photographs.

#### 3.6.4. Storage Stability

Ten milliliters of PEs were stored for 1 day (1 h), 5 days (120 h), 10 days (240 h), and 15 days (360 h) at room temperature, and the *CI* was measured by using Equation (3) [3].
(3)$$CI(\%)=\frac{H}{Ht}\times 100\%$$
where *H* and *H*_t_ represent the cream layer and total emulsion height, respectively.

### 3.7. Statistical Analysis

All experiments were repeated three times. The significance was determined by one-way ANOVA with SPSS 20 (SPSS version 20, SPSS Inc., Chicago, IL, USA), and *p* < 0.05 was used to determine the statistical significance of differences.

## 4. Conclusions

The nano-WalPI was prepared by using the US–pH-cycling method, and the PEs were produced using the nano-WalPI as an emulsifier. The average size of the nano-WalPI was 108.20 nm, which has good functional properties and could form a stable O/W type of PEs. The nano-WalPI concentration and oil volume fraction could affect the stability of PEs. The stability of PEs could be enhanced by increasing the nano-WalPI concentration from 0.5% to 2.0% (oil volume fraction, 20–70%) at a constant oil volume fraction (nano-WalPI concentration). The PEs were the most stable when the nano-WalPI concentration was 2.0% and the oil volume fraction was 60%. In brief, nano-WalPI may be utilized as a good emulsifier to prepare PEs at an appropriate nano-WalPI concentration and oil volume fraction, which provides a good basis for plant protein emulsion as a fat substitute.

## Figures and Tables

**Figure 1 molecules-28-05434-f001:**
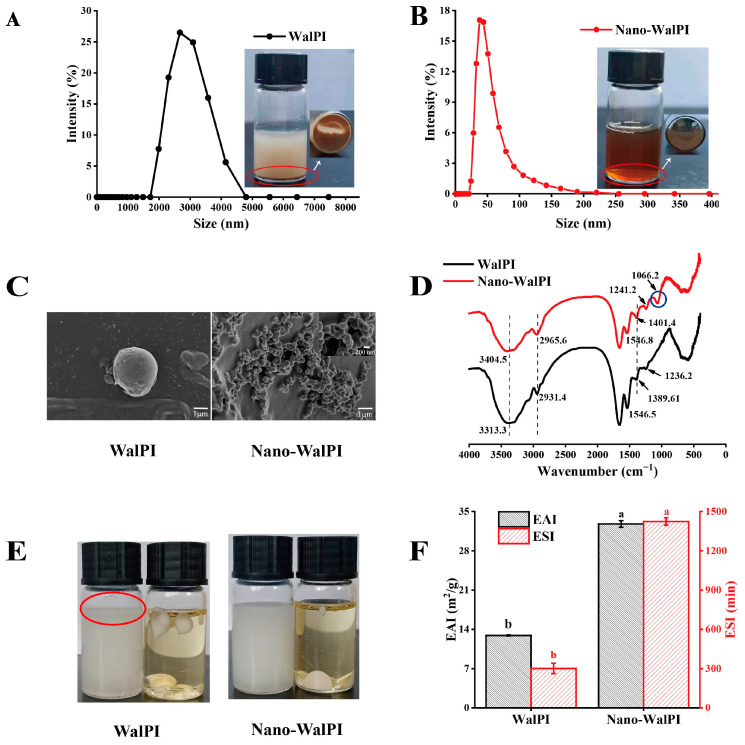
(**A**) Particle size distributions of walnut protein isolate (WalPI) and walnut protein isolate nanoparticles (nano-WalPI), (**B**) average size of WalPI and nano-WalPI, (**C**) SEM of WalPI and nano-WalPI, (**D**) FTIR spectra of WalPI and nano-WalPI, (**E**) the type of PEs of WalPI and nano-WalPI, and (**F**) the EAI and ESI of WalPI and nano-WalPI. The different lowercase letters (a–b) in Figure (**F**) indicate significant differences (*p* < 0.05). The red circle indicates the oil layer in Figure (**E**).

**Figure 2 molecules-28-05434-f002:**
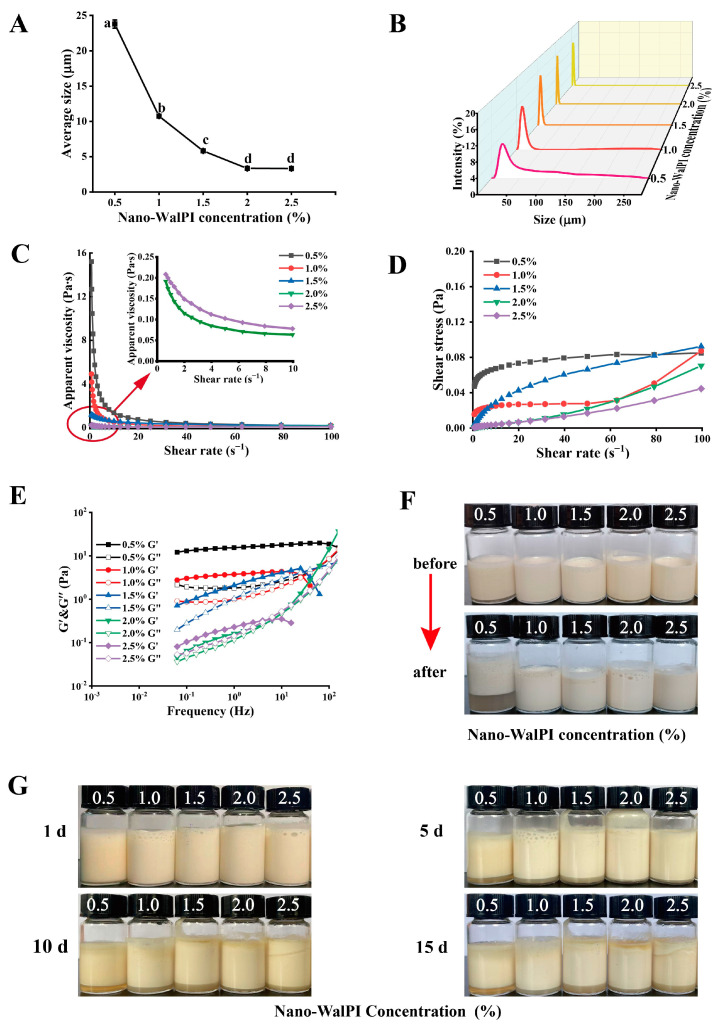
Effect of nano-WalPI concentrations on (**A**) the average size, (**B**) droplet size distributions, (**C**) apparent viscosity, (**D**) shear stress, (**E**) storage modulus (G′) and loss modulus (G″), (**F**) appearance before and after heating, and (**G**) appearance after storage of the PEs. The different lowercase letters (a–d) in Figure (**A**) indicate significant differences (*p* < 0.05).

**Figure 3 molecules-28-05434-f003:**
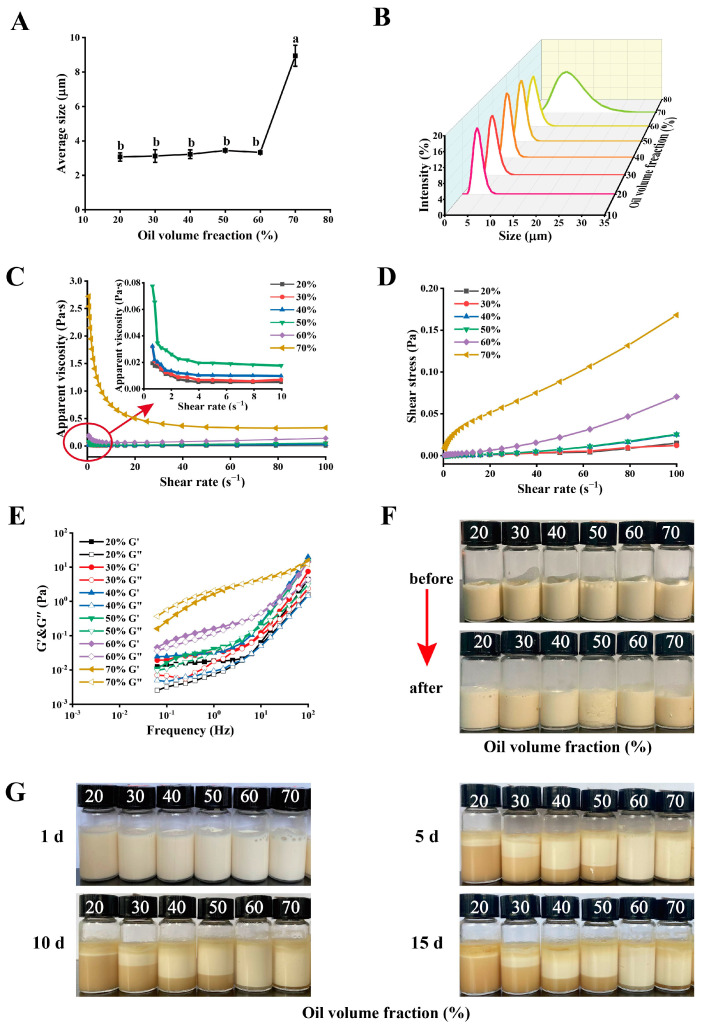
Effect of oil volume fractions on (**A**) the average size, (**B**) particle size distributions, (**C**) apparent viscosity (**D**) shear stress, (**E**) G′ and G″, (**F**) appearance before and after heating, and (**G**) appearance after storage of PEs. The different lowercase letters (a–b) in Figure (**A**) indicate significant differences (*p* < 0.05).

**Table 1 molecules-28-05434-t001:** The average size, PDI, and H_0_ of WalPI and nano-WalPI.

Sample	Average Size/nm	PDI	H_0_
WalPI	3368.83 ± 119.19 ^a^	0.56 ± 0.02 ^a^	875.21 ± 2.66 ^b^
Nano-WalPI	108.20 ± 2.89 ^b^	0.40 ± 0.09 ^b^	1027.43 ± 4.16 ^a^

Different letters indicate significant differences at *p* < 0.05.

**Table 2 molecules-28-05434-t002:** Secondary structure of WalPI and nano-WalPI.

Sample	α-Helix/%	β-Sheet/%	β-Turn/%	Random Coil/%
WalPI	18.08 ± 0.05 ^b^	53.67 ± 0.32 ^a^	13.97 ± 0.39 ^b^	14.34 ± 0.02 ^b^
Nano-WalPI	26.22 ± 0.36 ^a^	39.67 ± 0.35 ^b^	17.42 ± 0.25 ^a^	16.68 ± 0.20 ^a^

Different letters indicate significant differences at *p* < 0.05.

**Table 3 molecules-28-05434-t003:** CI values of the PEs at different nano-WalPI concentrations.

Nano-WalPI Concentration/%	CI/%			
	1 d (1 h)	5 d (120 h)	10 d (240 h)	15 d (360 h)
0.5	0 ^Ca^	11.75 ± 1.33 ^Ba^	13.16 ± 0.53 ^Ba^	18.07 ± 1.10 ^Aa^
1.0	0 ^Ba^	6.85 ± 1.40 ^Ab^	9.44 ± 0.55 ^Ac^	12.11 ± 1.39 ^Ac^
1.5	0 ^Ca^	4.60 ± 0.73 ^Bc^	11.29 ± 0.95 ^Ab^	15.35 ± 1.93 ^Ab^
2.0	0 ^Da^	1.00 ± 0. 25 ^Cd^	2.67 ± 0.29 ^Be^	4.67 ± 0.58 ^Ae^
2.5	0 ^Ca^	1.58 ± 0.53 ^Cd^	4.91 ± 0.61 ^Bd^	9.07 ± 0.85 ^Ad^

Statistically significant differences in the same rows are indicated by uppercase letters (*p* < 0.05), and significant differences within the same column are indicated by lowercase letters (*p* < 0.05), same as below.

**Table 4 molecules-28-05434-t004:** CI values of the PEs at a different oil volume fractions.

Oil Volume Fraction/%	CI/%			
	1 d (1 h)	5 d (120 h)	10 d (240 h)	15 d (360 h)
20	0 ^Ca^	84.91 ± 0.80 ^Ba^	90.88 ± 1.61 ^Aa^	91.05 ± 1.39 ^Aa^
30	0 ^Db^	59.65 ± 1.61 ^Cb^	64.39 ± 1.69 ^Bb^	73.16 ± 1.90 ^Ab^
40	0 ^Db^	24.92 ± 1.20 ^Bc^	30.00 ± 1.26 ^Ac^	30.48 ± 0.82 ^Ac^
50	0 ^Cb^	23.86 ± 0.30 ^Cc^	27.89 ± 1.82 ^Bc^	32.46 ± 1.85 ^Ac^
60	0 ^Db^	1.00 ± 0. 25 ^Cd^	2.67 ± 0.29 ^Bd^	4.67 ± 0.58 ^Ad^
70	0 ^Cb^	2.33 ± 0.29 ^Bd^	2.50 ± 0.50 ^Bd^	7.17 ± 1.04 ^Ad^

Statistically significant differences in the same rows are indicated by uppercase letters (*p* < 0.05), and significant differences within the same column are indicated by lowercase letters (*p* < 0.05).

## Data Availability

Not applicable.

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
