# Peer review of "Development and Characterization of Pickering Emulsion Stabilized by Walnut Protein Isolate Nanoparticles"

_molecules, 2023, doi:10.3390/molecules28145434_

Round 1
Reviewer 1 Report
It is an interesting article and it can be published in the journal after making the following corrections:
- Line 54; We all know WPI as whey protein isolate. It is better to use another abbreviation for walnut protein isolate (e.g., WalPI).
- Line 54; The logical reason for using ultrasound to prepare nanoparticles is not presented in the introduction section.
- Lines 94-95; How did the authors come to this conclusion? For this purpose, you must have two treatments: pH-cycled sample and Ul-treated sample. In fact, we do not know that this effect is related to the ultrasound rather than the pH change. Therefore, it is suggested to add pH-cycled sample and Ul-treated sample to the treatments.
- Line 111; …3.1.1… should be 2.1.1.
- Line 114; What is the effect of cavitation on protein structure? Please explain.
- Lines 119-134; The discussion of the FTIR section is very weak. Authors should explain the index peaks. In addition, which functional group does peak 1066 belong to? Visual examination of the spectra shows that apparently only the intensity of the peaks has changed and not the shift of the peak.
- Line 150; please change “hydrophily” to “hydrophobicity”.
- Lines 149-150; To reach such a result, the flexibility and surface hydrophobicity tests should be performed. Furthermore, is there always a strong correlation between these two parameters and improved emulsifying activity?
- Line 346; Why is GDL used to adjust the pH to 7.0? Why is HCl not used for this purpose?
- Lines 347-349; How to ensure that the nanoparticle is in the supernatant phase and not in the sediment phase?
Reviewer 2 Report
In this study the authors fabricated a new Pickering emulsion using walnut protein isolate nanoparticles (nano-WPI). They comprehensively characterized physiochemical properties of this nano-WPI, including particle size, microstructure, and emulsion properties. The authors further examine the droplet size, rheological properties, thermal stability, and storage stability of Pickering emulsion made by nano-WPI. The findings in this study expand the mechanistic study of Pickering emulsion and have broad implications in food-grade Pickering emulsion stabilizers.
Generally, the work presented is with great quality and significant interest. It is suitable for publication in Molecules after minor revisions in scientific writing and data interpretation. It’s expected this could be an impactful paper after these improvements. Below are a few comments:
1) What is the general yield of preparing nano-WPI from WPI? What is the dialysis membrane cutoff used for dialysis? Please provide relevant information so that it allows reproducible work for following studies.
2) Could authors elaborate further what happen during the treatment of ultrasonication combined with pH-cycling method (UL–pH-cycling method)? Is there any chemical reaction happening? In other words, does the chemical structure of components in WPI change?
3) Page 6, line 181: It seems apparent viscosity in 2.5% nano-WPI is higher than that in 2%, which is inconsistent with the trend observed in other concentrations. Do authors have any hypothesis? Please clarify.
Please double check the grammar and also make sure there is no typos in the updated version of manuscript.
Reviewer 3 Report
This manuscript aimed to develop Pickering emulsions stabilized by walnut protein isolate nanoparticles (nano-WPI), and to assess the impact of nano-WPI concentrations and oil volume fractions on the physico-chemical and rheological properties and stability of the system. Overall, it is well designed study, methods are appropriate (although some suggestions are given later), and the obtained results bring novelty to this research field. However, there are several (both major and minor) concerns in the manuscript that need to be clarified.
Abstract: What range of nano-WPI concentrations and oil volume fractions was investigated? It should be defined here.
Lines 28/29: You should be more specific; this does not apply to all Pickering emulsions. It depends on what kind of solid particles are used.
Line 30: “can be” instead of “contain”.
Lines 36/37: it should be clear that protein emulsions are O/W most often.
Line 53: “PEs can be prepared as emulsifiers” – incorrect sentence, it must be modified.
Line 55: Why UL and not US?
Line 60: Emulsions are always thermodynamically unstable systems, but nano-proteins can stabilize them (not prevent formation of unstable systems). Please, rephrase.
Lines 69/70: It should be clear (both here and later in the text) that it refers to creaming stability. How is it explained that larger droplets lead to better stability?
Line 76: What do you mean by studying the “functional group”?
Line 80: “in the food area” – please, rephrase.
Line 82: It should be Results and Discussion.
Lines 87/88: “a single peak with a monodisperse system” – not clear, a single peak that indicates the existence of a monodisperse system.
Lines 91 and 92: “at the bottom” should be deleted; precipitation is always at the bottom.
Line 101: “The results reduced the average size” – “results” should be changed.
Line 111: It should be 2.1.1.
Line 112: Please explain further what you mean by uniform distribution of hydroniums.
2.1.3. FTIR analysis section: you need to refer to Table 2 in the text. Define a and b in the table as well.
Line 148: It is better to state how many times EAI and ESI were higher when nano-WPI was used (instead of using percentages).
Lines 149/150: When stating that “the unfolding of the protein structure improved the flexibility and surface hydrophily of the nano-WPI”, it would be good if it was confirmed by surface hydrophobicity test (with ANS for example). Besides, the term used should be “hydrophilicity”.
Figure 2 legend: “shear stress” should be instead of “stress”, and G’ and G’’ should be defined here.
Line 181: “increased” instead of “enhanced”.
When increasing the concentrations of nano-WPI, the emulsions showed Newtonian fluid behavior at lower shear rates. Did other authors observe similar effects for PEs? It should be further discussed.
In oscillatory measurements, emulsion with 1.5% of nano-WPI exerted the properties of a liquid with viscous deformation (G′ < G″), while it was not the case at lower or higher concentrations (where the formation of gel structure occurred). How is it possible?
Section 2.3. – It should be clear that the nano-WPI concentration was 2%. Besides, explain further why this concentration was chosen.
Section 2.3.1. – refer to Figure 3 in the text. Again, “increased” should be instead of “enhanced”.
The next parts of the manuscript are contradictory:
Line 158: You state that “The average size and droplet size distribution of the PEs are indicators of the stability of emulsions”. The droplet size for 70% oil emulsions was significantly higher than for those with lower oil concentrations. Lines 258/259: “nano-WPI is not enough to cover the area, which causes droplets to collide and aggregate”. Lines 297/298: “Rising oil volume fractions (20%–60%) could improve the gel network structure and stability of PEs”. Only at 70% oil, G’’ was dominant at lower frequencies. Lines 324-326: “Li et al. [19] showed that raising the oil volume fraction could improve the average size, apparent viscosity, storage modulus, and stability of PEs, which was compatible with the findings of this paper.” Creaming stability for 70% oil is much lower than for 20-50% (Table 4), which does not comply with the discussion of droplet sizes. This whole part (regarding the impact of oil volume fraction) needs to be modified when discussing the physical properties, rheological behavior and stability of PEs together.
Line 333: Define GDL here in the text.
Lines 343: Was it dissolved or dispersed?
Line 344: “the mixed solution was dispersed” – it is not clear, how was the solution dispersed?
Section 3.4.4: Why didn’t you perform conductometric analysis?
Line 374: obtained?
Define N, C and phi in the equations.
Line 382: And 20%?
Line 383: I find the information on the prepared volume of emulsion redundant.
Section 3.6.2: It should be clarified. It should be shear stress instead of just stress. Besides, how did you select the frequency range and the shear stress applied for the oscillatory measurements? Did you perform amplitude sweep tests in order to determine the linear viscoelastic region (LVR) by recording the storage (G’) and loss (G”) moduli versus shear stress at a constant frequency?
Section 3.7: Which ANOVA post hoc tests were conducted?
Eventually, the conclusions shouldn’t be the results repetition (with statistical significance etc), but the brief answer to the objectives of the study that were set. This section needs to be rephrased.
There are some major concerns regarding the use of language. Detailed language check by a native speaker needs to be done.
Round 2
Reviewer 1 Report
Accept in present form
Reviewer 3 Report
The authors have adopted all of the suggestions raised.
There are still some language errors that should be corrected.